# Human IL-2Rα subunit binding modulation of IL-2 through a decline in electrostatic interactions: A computational and experimental approach

**Arezoo Beig Parikhani**[1,2‡], **Kowsar Bagherzadeh**[3,4], **Rada Dehghan**[1,2‡], **Alireza Biglari**[5], **Mohammad Ali Shokrgozar**[6], **Farhad Riazi Rad**[7], **Sirous Zeinali**[8], **Yeganeh Talebkhan**[9], **Soheila Ajdary**[7]*, **Reza Ahangari Cohan**[10]*, **Mahdi Behdani**[1]*

1 Venom and Biotherapeutics Molecules Laboratory, Department of Medical Biotechnology, Biotechnology Research Center, Pasteur Institute of Iran, Tehran, Iran, 2 Student Research Committee, Pasteur Institute of Iran, Tehran, Iran, 3 Stem Cell and Regenerative Medicine Research Center, Iran University of Medical Sciences, Tehran, Iran, 4 Eye Research Center, The Five Senses Health Institute, Rassoul Akram Hospital, Iran University of Medical Sciences, Tehran, Iran, 5 School of Medicine, Tehran University of Medical Sciences, Tehran, Iran, 6 National Cell Bank of Iran, Pasteur Institute of Iran, Tehran, Iran, 7 Department of Immunology, Pasteur Institute of Iran, Tehran, Iran, 8 Molecular Medicine Department, Biotechnology Research Center, Pasteur Institute of Iran, Tehran, Iran, 9 Department of Medical Biotechnology, Biotechnology Research Center, Pasteur Institute of Iran, Tehran, Iran, 10 Department of Nanobiotechnology, New Technologies Research Group, Pasteur Institute of Iran, Tehran, Iran

‡ ABP and RD contributed equally to this work as co-first authors
* behdani73042@yahoo.com, behdani@pasteur.ac.ir (MB); cohan_r@yahoo.com (RAC); ajdsoh@pasteur.ac.ir (SA)

## Abstract

Although high-dose IL-2 has clear antitumor effects, severe side effects like severe toxicity and activation of Tregs by binding of IL-2 to high-affinity IL-2R, hypotension, and vascular leak syndrome limit its applications as a therapeutic antitumor agent. Here in this study, a rational computational approach was employed to develop and design novel triple-mutant IL-2 variants with the aim of improving IL-2-based immunotherapy. The affinity of the mutants towards IL-2Rα was further computed with the aid of molecular dynamic simulations and umbrella sampling techniques and the obtained results were compared to those of wild-type IL-2. *In vitro* experiments by flow cytometry showed that the anti-CD25 mAb was able to bind to PBMC cells even after mutant 2 preincubation, however, the binding strength of the mutant to α-subunit was less than of wtIL-2. Additionally, reduction of IL-2Rα subunit affinity did not significantly disturb IL-2/IL2Rβγc subunits interactions.

## 1. Introduction

Human Interleukin-2 (IL-2) is a pleiotropic cytokine that plays pivotal roles in immune responses [1]. The protein promotes the proliferation, differentiation, and survival of T and B cells, and also enhances the cytolytic activity of natural killer (NK) cells in the innate immune

**Data Availability Statement:** All relevant data are within the manuscript and its Supporting Information files.

**Funding:** This study is supported by a PhD grant from Pasteur Institute of Iran and by grant No: 980902 from the Biotechnology Development Council of the Islamic Republic of Iran to Arezoo Beig Parikhani. The funders had no role in study design, data collection and analysis, decision to publish, or preparation of the manuscript.

**Competing interests:** The authors have declared that no competing interests exist.

**Abbreviations:** ConA, Concanavalin A; DSSP, Secondary structure analysis; FDA, Food and Drug Administration; HD-IL-2, High-dose human interleukin-2; IL-2, Human interleukin-2; IL-2R, Interleukin-2 receptor; IFN, Interferon; IPTG, Isopropyl β-D-1-thiogalactoside; M1, Mutant 1 human interleukin-2; M2, Mutant 2 human interleukin-2; Mab, Monoclonal antibodies; MDs, Molecular dynamic Simulations; PBMC, Peripheral blood mononuclear cells; rIL-2, Recombinant wild-type human interleukin 2; RMSD, Root mean square deviations; RMSF, Root mean square fluctuations; SDS-PAGE, Sodium dodecyl sulphate–polyacrylamide gel electrophoresis; TNF, Tumor necrosis factor; Treg, Regulatory T cells; VLS, Vascular leak syndrome; WHAM, Weighted histogram analysis method; wtIL-2, Wild-type human interleukin-2.

defense [2]. IL-2 is a 15-kDa glycoprotein produced primarily by activated CD4+ and CD8+ T lymphocytes [3]. This cytokine exerts its effect by binding to a heterotrimeric receptor on the surface of immune cells. The high affinity IL-2 receptor is composed of three separate, noncovalently linked subunits, termed IL-2Rα (CD25), IL-2Rβ (CD122), and IL-2Rγc (CD132) [4]. The IL-2Rα chain captures IL-2 at the cell surface and delivers it to IL-2Rβγc chains; the signaling part of the receptor [4]. The β and γ chains are expressed on T cells while the α chain expression is restricted to early thymocytes, Tregs, and activated T cells. While β and γ chains together form an IL-2 receptor (IL-2R) with intermediate affinity, the α chain is unable to form a functional receptor in the absence of βγc chains [4].

High-dose (HD) IL-2 was approved by the FDA for the treatment of metastatic renal cell carcinoma in 1992 and metastatic melanoma in 1998 [5]. IL-2 has also been used for the treatment of leukemias and lymphomas [6]. Based on the administered dose of IL-2, it can act as a promoter of both immunosuppression via Tregs, and immune stimulation via other CD4+, CD8+ T, and NK cells [7]. This dual effect of IL-2 on the immune response limits its applications as a therapeutic antitumor agent [8]. The expression level of IL-2Rα on resting cytotoxic lymphocytes, such as NK and CD8+ T cells, is low or undetectable, so these cells are not activated by low-doses of IL-2 [9]. IL-2Rα expression on these cells increases after the initial stimulation and is required for maximal lymphocyte proliferation. While, high affinity IL-2R on Tregs can compete more effectively for IL-2 at low levels, HD-IL-2 can activate even resting cytotoxic lymphocytes and it is used for immune-stimulatory and antitumor activity [9].

HD-IL-2 administration has been associated with life-threatening toxicities such as vascular leak syndrome (VLS) and pulmonary edema [10], so most patients do not benefit from HD-IL-2 therapy [9]. Lowering IL-2 dose could limit side effects but also decreases the efficacy [11]. It is reported that part of the limitation in IL-2 efficacy for cancer treatment is related to IL-2-driven expansion of Tregs, which leads to a reduced antitumor immunity [12]. An increased level of Tregs has been found in most cancer patients and in cases such as breast cancer, renal cell carcinoma, and non-small cell lung cancer, the increment has been associated with worse disease outcomes or poor survival [13].

Many attempts have been made to improve efficacy of IL-2-based therapy and enhance the safety profile for patients, including; changes in route of administration and combinations with other drugs [3]. However, the issues related to the toxicity and efficacy still remain. Molecular strategies to generate modified forms of IL-2 could be helpful in these areas [7,11]. Therefore, in the current study, we designed and tested a novel triple-mutant IL-2 variant with a reduced affinity to the IL-2Rα subunit using a computational approach and experimental analysis.

## 2. Materials and methods

### 2.1. Computational studies

The schematic representation of the computational procedure performed in this study is illustrated in S1 Fig.

**2.1.1. Design and interaction analysis of IL-2 variants.** The X-ray structure of the complex between IL-2 and IL-2 receptor (PDB ID: 2b5i) was retrieved from the protein data bank (https://www.rcsb.org). The PDB file was then submitted to PDBsum in the generate mode and the residues involved in hIL2-hIL2Rα electrostatic interactions were identified [14]. Then, alanine mutations were applied to the crystal structure of the hIL2-IL-2Rαβγc complex (PDB ID: 2b5i) at the selected positions using Swiss-Pdb Viewer v4.1 [15]. The affinity between IL-2 mutants and IL-2R α-subunit was computed by PRODIGY (https://nestor.science.uu.nl/prodigy/) and PDBePISA (https://www.ebi.ac.uk/pdbe/pisa/) webservers.

**2.1.2. Docking procedure.**   To predict the best interacting model between IL-2 and the receptor, calculating the approximate binding energy, and obtaining an appropriate initial structure to perform molecular dynamic (MD) simulations, the energy minimized 3D-models were docked over IL-2Rαβγc using ClusPro webserver [16]. Briefly, all hetero-atoms including waters, ligands, and cofactors were removed. Then, IL-2 and IL-2Rαβγc were submitted as ligand and receptor, respectively. Finally, the analysis was carried out by defining attraction residues. The docking procedure generated a number of detailed models that were sorted by the cluster size. Finally, the best cluster was chosen according to the predicted binding energies and modes of interactions for further analysis.

**2.1.3. Simulations procedure.**   MD simulations were carried out using GROMACS v5.1.5 software [17] with GROMOS AMBER force field (amber99sb-ildn) [18] on the native and mutated IL-2 [mutant 1 (M1) and mutant 2 (M2)] as well as M1 and M2 structures in complex with IL-2Rαβγc, for the best-ranked models obtained from the molecular docking studies. The models were solvated in a dodecahedral box of TIP3P water molecules [19] with a minimum distance of 14 Å between the protein surface and the box wall. The net charge of the system was neutralized by replacing water molecules with appropriate counter sodium and chloride ions. The van der Waals cutoff was set to 14 Å. Periodic boundary conditions were assigned in all directions. The solvated system was then minimized through the steepest descent algorithm [20] with 1000 KJ mol$^{-1}$ nm$^{-1}$ tolerance followed by a canonical ensemble (NVT) and isothermal-isobaric ensemble (NPT) for 20 ps. The temperature and pressure of the system were independently maintained using Berendsen thermostat and Parrinello-Rahman barostat algorithm [21] at constant temperature and pressure of 300 K and 1 bar, respectively. The particle mesh Ewald (PME) algorithm was employed to calculate the long-range electrostatic interactions [22]. The LINCS algorithm [23] was applied to restrain all the bonds with an integration step of 1 fs. Finally, the whole system was subjected to 100 ns of MDs at constant pressure and temperature. The stability of the computed structures was investigated by calculating the root mean square deviations (RMSD) and root mean square fluctuations (RMSF) during the simulation. The coordinate files were finally extracted from the trajectories for further analysis.

**2.1.4. Umbrella sampling procedure.**   An umbrella sampling algorithm was employed to calculated potentials of mean force (PMF) [24]. The average structure from the last 40 ns of each simulation was selected for the umbrella sampling study to calculate the potential of mean force (PMF) for the native and mutant complexes. The already equilibrated complexes were first made parallel to the z-axis. A constant velocity pulling procedure with a rate of 1 nm was assigned to pull IL-2Rα along the z-axis by a 6.0 nm distance. Therefore, a box was constructed with a z-axis length of 12 nm. The prepared box was solvated, neutralized, minimized, and equilibrated at a temperature (NVT) and specific pressure (NPT) similar to the previous MD simulations. The umbrella sampling simulation began with the center-of-mass-pulling method. IL-2Rα was pulled from the complex towards the solvent bulk over the course of 500 ps by using a 1000 kJ/(mol*nm) force, at the rate of 0.01 nm per ps. During this simulation, snapshots were saved at each 10 ps, so in total 50 configurations were generated from the pulling simulations. Eventually 22 to 23 configurations with a spacing of 0.2 nm were obtained for each complex to ensuring sufficient overlap of the probability distribution of each configuration. The selected configurations were then used as the starting configurations for each umbrella sampling simulation and were subjected to 1 ns of MDs after a brief NPT equilibration, independently. The potential mean force (PMF) was calculated using the weighted histogram analysis method (WHAM) [25]. The binding free energy (ΔG) was calculated for each system by taking the difference between the plateau region of the PMF curve and the energy minimum of each simulation.

**2.1.5. Data analysis and graphical presentation.** The MDs results were analyzed and visualized with Schrödinger and Pymol packages [26,27]. The graphs were all represented visually by using Microsoft Office Excel.

## 2.2. Experimental studies

**2.2.1. Protein expression, identification, and purification.** For the expression of native and mutant IL-2 (M2), the synthetic genes encoding the wild and M2 IL-2 were subcloned separately into pET28a expression vector using *Nco*I and *Hind*III restriction sites. The expression vectors were transformed into *E. coli* BL21 (DE3) strain using the heat shock method. Transformed clones were inoculated into 200 ml Luria Broth (LB) medium supplemented with 50 μg/mL kanamycin. The protein expressions were induced with 0.5 mM isopropyl β-D-1-thiogalactoside (IPTG) (Sigma, USA) at an OD 600nm of 0.5 and the bacterial pellet was collected 6 hours post-induction by centrifugation at 6000 rpm for 20 minutes and subjected to 12% SDS-PAGE to analyze the protein expression.

For western blotting, the proteins were transferred onto nitrocellulose membrane using Semi-Dry Transfer system (Bio-Rad, USA) for 45 min at 200 mA. The membrane was blocked in 3% skimmed milk in PBS for 2 h at room temperature and then incubated overnight with 1:2000 dilution of rabbit anti-Histidine primary antibody at room temperature. After 3 times washing with PBS-tween20 0.05%, the membrane was incubated with 1:2000 goat anti-rabbit HRP-conjugated secondary antibody (Sigma, USA) for 4 h. Finally, the membrane was washed 3 times similar to the previous step and stained using 3,3'-diaminobenzidine (DAB) substrate.

For protein purification, the bacterial pellets were resuspended in lysis buffer (10 mM imidazole, 0.5 M NaCl, 50 mM $NaH_2PO_4$; pH8.0) and sonicated at an amplitude of 100% (30-sec pulses with 10-sec intervals, for 10 min). The lysates were then centrifuged at 10,000 g for 20 min. The pellet, containing inclusion bodies (IBs), was solubilized in solubilization buffer (8M urea, 50 mM $NaH_2PO_4$, 300 mM NaCl, 10 mM imidazole) and placed under agitation for 1h at room temperature. After centrifugation (10,000 g for 30 min), the supernatant was filtered by 0.45 μm filters and applied to Ni-NTA resin at a low flow rate (1 ml/min). The recombinant protein was refolded on the column through a gradual removal of urea (from 8M to 0) by the refolding buffer (50 mM $NaH_2PO_4$, 0.5 M NaCl, 20 mM Imidazole; pH 8.0) over a period of 2 h. The His6-tagged proteins were eluted with the elution buffer (250 mM imidazole, 50 mM $NaH_2PO_4$, 0.5 M NaCl; pH 8.0). The purity of refolded soluble elution fractions was analyzed using 12% SDS-PAGE.

**2.2.2. PBMC isolation, stimulation, and flow cytometry.** Human peripheral blood mononuclear cells (PBMC) were separated from a healthy blood donor using Ficoll-Hypaque (Lymphodex, Innotrain, Germany) by density gradient centrifugation at 400 g at room temperature for 30 min according to the manufacturer's protocol. The cells were washed three times with sterile phosphate-buffered saline (PBS), counted, and resuspended in RPMI medium supplemented with 10% fetal bovine serum (FBS), 2 mM L-glutamine, 100 U penicillin, and 100 μg streptomycin defined as complete medium (CM). For the induction of CD25 expression, PBMCs were stimulated with 5 μg/ml concanavalin A (ConA) for 24 h. After washing with PBS, the cells were incubated with IL-2 or IL-2 mutein in CM for 24 h. His-tagged recombinant proteins bound-cells were detected with rabbit anti-His-tag Ab and mouse anti-rabbit FITC-conjugated. In another experiment, the cells previously incubated with wtIL-2 and M2, were stained with anti-CD25 MAb-PE antibody (eBioscience, United States). After adequate washing, flow cytometry was performed using Partec PAS III flow cytometer (Partec GmbH, Gorlitz, Germany), and data were analyzed by FlowJo software (Tree Star. Inc., Ashland, OR, USA). Isotype-matched control antibodies were used to detect non-specific binding to the cells.

## 3. Results

### 3.1. Computational studies

**3.1.1. Residue identifications.** The overall structure of IL-2 is presented in Fig 1A. Analyzing the interactions between IL-2 and IL-2Rα revealed that residues K35, R38, F42, K43, Y45, E61, E62, P65, L72, and Y107 are involved in the electrostatic interactions with IL-2Rα. While residues E61, E62, P65 and L72 and Y107 are located in a loop structure (A-B loop and C-D loop, respectively), the other target residues are constituents of β sheets and α helixes. To generate the initial structure of IL-2/IL-2R complex, the target residues were individually mutated to alanine using Swiss-Pdb Viewer v4.1 (15). The affinity between IL-2 mutants and IL-2Rα subunit was evaluated with the aid of PRODIGY (https://nestor.science.uu.nl/prodigy/) and PDBePISA (https://www.ebi.ac.uk/pdbe/pisa/) web servers. By keeping default parameters, the dissociation constants ($K_d$) and binding affinity ($\Delta G$) were finally calculated at 25˚C. Based on the obtained *in silico* affinity measurements, three mutations that had more impact on the affinity reduction of IL-2Rα subunit were selected (S1 Table). The results obtained from PRODIGY showed that 8 out of 9 single amino-acid substitutions increased binding energy relative to wtIL-2, among which K35A, F42A, and E61A resulted in higher binding free energies and eventually larger reduction in binding affinity for IL-2Rα in comparison to other mutations and wtIL-2. Also, PISA analysis revealed that F42A, P65A, and L72A would lower the binding affinity for IL-2Rα in comparison to the rest of the assigned mutations as well as wtIL-2. Therefore, two variants of triple mutant F42A, P65A, and L72A as mutant 1 (M1) and K35A, F42A, and E61A as mutant 2 (M2) were designed.

**3.1.2. *In Silico* characterization of the designed variants.** The designed variants as well as the wtIL-2 were submitted to a comprehensive set of atomistic MD simulations to investigate how the assigned mutations would alter the native structure of the target protein, IL-2, and predict appropriate conformations of the variants for predicting their interactions with IL-2Rαβc. The obtained results were analyzed as the time evolution in detail for the native and mutant conformations. Basic MD simulation trajectory analysis, including root mean square deviation (RMSD), root mean square fluctuation (RMSF), and the secondary structure analysis (DSSP) were performed for wtIL-2 as well as M1 and M2 as the time evolution of the native and mutant conformations using GROMACS software.

The backbone RMSD plots of the target structures showed that wtIL-2 and M2 trajectories become stable after almost 38 ns and 55 ns of simulations, respectively, while for that of M1, it becomes stable after 45 ns of simulations and stays flat up to nanosecond 80 ns and then decrease and stay stable for the 10 ending nanoseconds of simulation time (Fig 1B). Totally, the root mean structure deviations of the target structures are less than 0.2 nm which shows probably no significant structural rearrangements like refolding of a helix into a loop or a β-sheet to a turn has occurred.

The per-residue root mean-square fluctuation (RMSF) plots of the wild and mutated types of IL-2 are presented in Fig 1C. The obtained results revealed that substitution of K35A decreased fluctuation in α helix for M2, while F42A and E61A behave like those in wtIL-2 structure (Fig 1D and 1E). In contrast, F42A, and L72A showed an increase in A-B loop and B helix fluctuations in comparison to wtIL-2 and M2 that can be attributed to the loss of π-π stack electrostatic interactions between P65, F42 and F43 (Fig 1F) and polar interactions between L72 and R38 (Fig 1G) due to alanine substitution. Also, the RMSF plots elucidated that M1 and M2 show fluctuation behaviors close to that of wtIL-2 which reveals the structural stability of the two mutants relative to the wtIL-2.

The structure of the wild type and mutants were further assessed employing secondary structure analysis (Fig 2). The obtained results show no significant refolding in the percent of

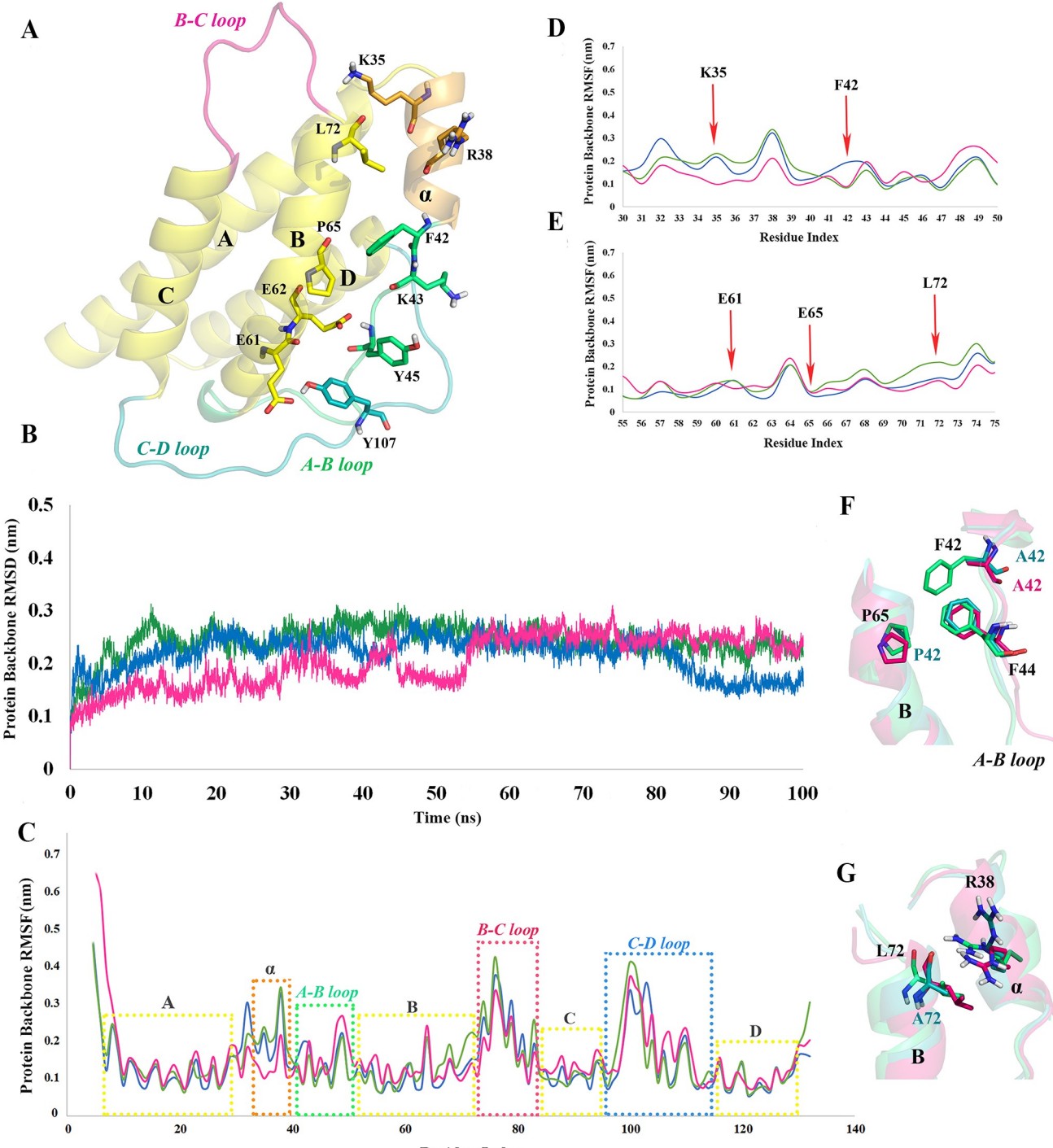

**Fig 1. A.** IL-2 protein structure (hot spot residues in sticks), **B.** Back-bone root mean square fluctuations (RMSD) of the native (green), mutant 1 (blue) and mutant 2 (magenta) protein, **C.** Per-residue root mean square fluctuations (RMSF) of the native (green), mutant 1 (blue) and mutant 2 (magenta) protein residues, **D.** Per-residue root mean square fluctuations (RMSF) of the target residues 30–50 close view, **E.** Per-residue root mean square fluctuations (RMSF) of the target residues 55–75 close view, **F.** Residues F42, F43 and P65, and **g.** Residues L72 and R38.

residues participating different structural arrangements (Fig 2A–2C). According to Fig 2D–2F, the assigned mutations has decreased the number of α-helix participating residues from 55%

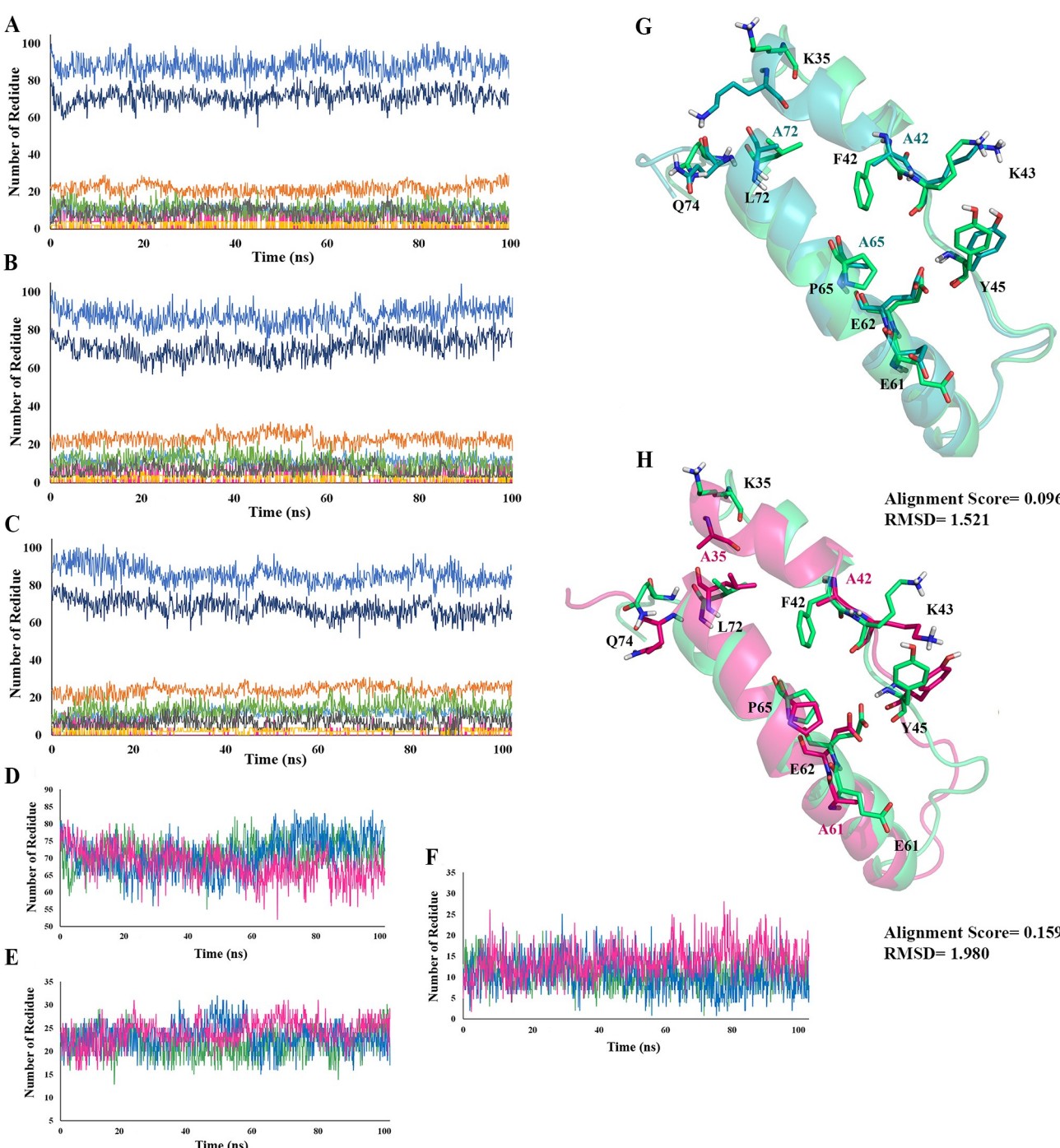

**Fig 2. Secondary structure analysis. A.** The native protein, **B.** Mutant 1 and, **C.** Mutant 2, during 100 ns of MDs (where the structure is shown in blue, α-helix in dark blue, 3-helix in gray, coil in orange, turn in green, β-sheet in magenta, b-bridge in yellow, and bend in cyan), **D.** α-helix structure of native structure (green), mutant 1(blue) and mutant 2 (magenta), **E.** Coil structure of native structure (green), mutant 1(blue) and mutant 2 (magenta), **F.** Turn structure of native structure (green), mutant 1(blue) and mutant 2 (magenta), **G.** superimposition of the native IL-2 and IL-2Rα interacting residues over mutant 1, and **H.** superimposition of mutant 2 and IL-2Rα interacting residues over mutant 2 IL-2.

in wtIL-2 and M1 to 53% in M2, and increased the number of the coil and turn participating residues from 8% and 18% in wtIL-2 and M1to 11% and 19% in M2, respectively. Additionally,

the average structures of the computed mutants were obtained from the plateau of the MD simulation and aligned over that of wtIL-2 (Fig 2G and 2H) and alignment scores of 0.096 and 0.159 and also RMSD values of 1.521 and 1.980 Å were obtained for M1 and M2 over wtIL-2, respectively.

Ramachandran Plot server (https://zlab.umassmed.edu/bu/rama/) was used to evaluate the reliability of the modeled mutants and as an energetic visualization of allowed and disallowed dihedral angles psi (ψ) and phi (φ) of amino acids (S2 Fig). The obtained results showed that the percent of the residues in the favored region are 95.33%, 95.16%, and 94.81% as well as 3.80%, 3.80%, and 4.15% in the allowed region for wtIL-2, M1 and M2, respectively, after 100 ns of simulation (S2 Table). Also, the number of residues in the disallowed region are obtained to be 0.865%, 1.036%, and 1.038% for wtIL-2, M1, and M2, respectively. Ramachandran plots demonstrated that the 3D modeled structures of the mutants represent favorable features, indicating the total percentage of favored and allowed region residues are more than 98%. Since the presence of more than 90% of the residues in the favored and allowed regions of the plot are considered ideal, the quality of the overall structures was desirable enough to be used for the upcoming computational studies [28].

**3.1.3. Docking studies.** Docking studies were conducted to predict the best interacting model between IL-2 and IL-2Rαβγc, calculate an approximate binding energy, as well as obtaining an appropriate mode of interactions for MD simulations. Therefore, the average structures of the computed wt-IL2 and mutants obtained from 100 ns of simulations were used as the structure to be docked over IL-2Rαβγc with the aid of ClusPro webserver (16). The interacting residues were assigned to the webserver and the best-ranked models of complexes were selected considering the type of interactions (mainly hydrogen bonds, hydrophobic and electrostatic interactions), the interacting residues, orientations, distance and the binding energies. The binding affinity (the dissociation constant, $K_d$) of the obtained complexes were predicted using PRODIGY webserver (Table 1). It is observed that R38A, F42A, and E61A substitutions considerably decrease the affinity of M2 for IL-2Rα in comparison to wtIL-2 and M1. To ensure that the decrease in the affinity is a result of a decline in the target residues interactions with IL-2Rα, wtIL-2 and the variants affinity for IL-2Rβ and IL-2Rγc were also studied by PRODIGY (S3 Table). The obtained results showed that the affinity of M2 for IL-2Rα decreases while increases the affinity of the mutant for IL-2Rβ. The assigned mutations affect the affinity of the M2 for IL-2Rγc. Since the IL-2Rγc binding site is close to that of IL-2Rα, mutations in the latter would probably induce conformational changes in the former site.

**3.1.4. Molecular dynamic simulations of the complex structures.** The top rank complexes obtained from docking studies were subjected to molecular dynamic simulations for 100 ns to check the stability of the docked structures as well as the observed interactions. The backbone root mean square deviations were computed individually for IL-2 and the variates as well as IL-2Rα, IL-2Rβ, and IL-2Rγc and are presented in Fig 3. RMSD plots of IL-2Rβ (Fig 3A) and IL-2Rγc (Fig 3B) reach the plateau after 15 ns and 5 ns, respectively, with RMSD values less than 0.1 nm, which shows the stability of their structures as well as their interactions with IL-2 and mutants M1 and M2. These phenomena further confirm that the alternations observed in the docking binding energies are due to the made mutations. Considering the RMSD plot of IL-2Rα (Fig 3C), the plot became stable after 30 ns of MDs and stayed steady

**Table 1. The binding and docking energy comparison between wtIL-2, M1, and M2 complexes.**

| | wtIL-2 | M1 | M2 |
|---|---|---|---|
| **Cluspro Docking (kcal/mol)** | -3935.1 | -3510.7 | -3410.9 |
| **The Prodigy analysis (the dissociation constant ($K_d$) M** | $2.6 \times 10^{-8}$ | $2.6 \times 10^{-8}$ | $1.2 \times 10^{-7}$ |

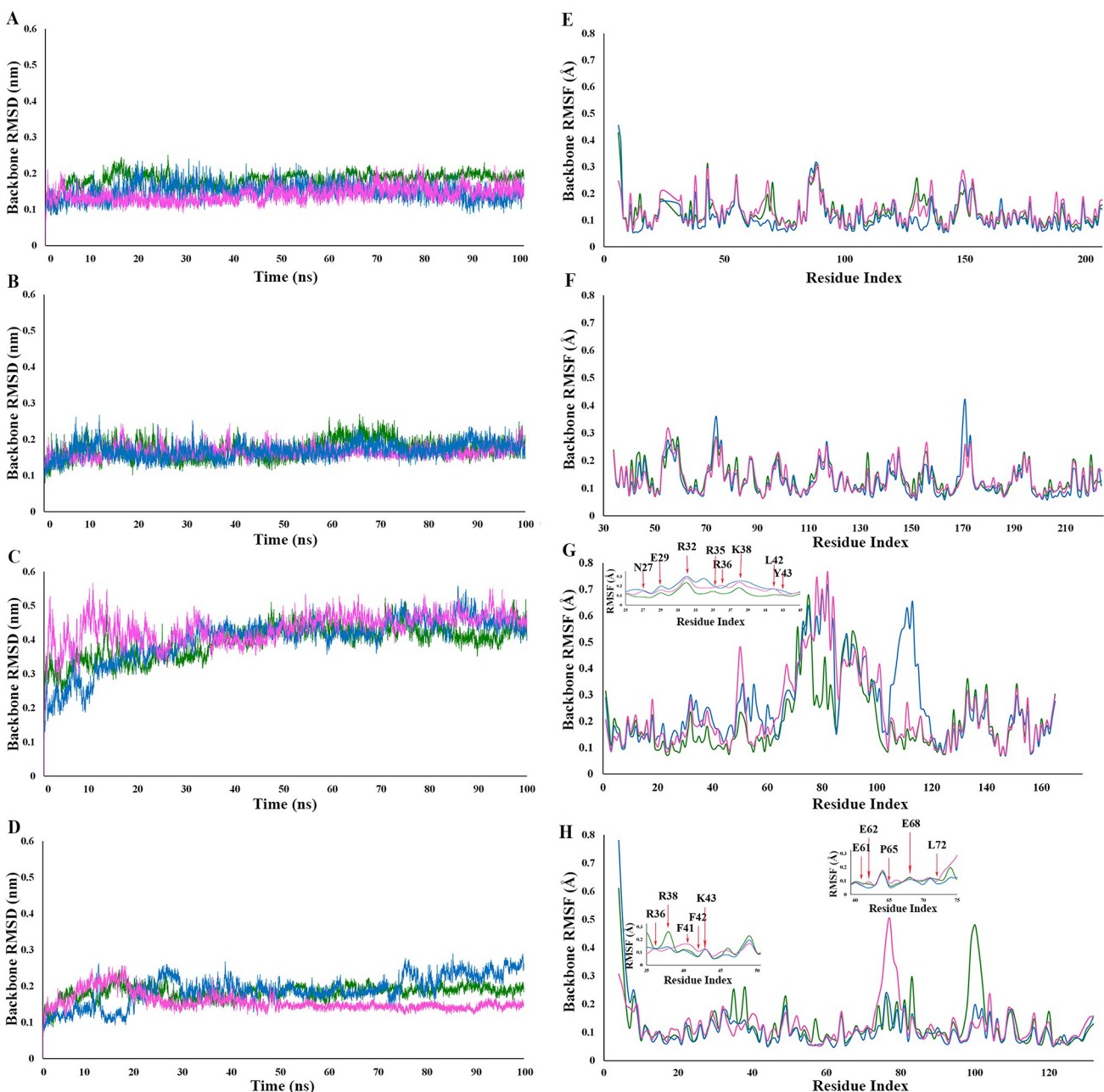

**Fig 3. Back-bone root mean square fluctuations (RMSD). A.** IL-2Rβ (the native (green), mutant 1 (blue) and mutant 2 (magenta) proteins), **B.** IL-2Rγc (the native (green), mutant 1 (blue) and mutant 2 (magenta) proteins), **C.** IL-2Rα (the native (green), mutant 1 (blue) and mutant 2 (magenta) proteins), and **D.** Native IL-2 (green), mutant 1 (blue) and mutant 2 (magenta) proteins during 100ns of MDs. Back-bone root mean square fluctuations (RMSF) of, **E.** IL-2Rβ residues (the native (green), mutant 1 (blue) and mutant 2 (magenta) proteins, **F.** γc residues (the native (green), mutant 1 (blue) and mutant 2 (magenta) proteins), **G.** IL-2Rα residues (the native (green), mutant 1 (blue) and mutant 2 (magenta) proteins), and **H.** wtIL-2 residues (green), mutant 1 (blue) and mutant 2 (magenta) during 100ns of MDs.

during the rest of the simulation. While the RMSD value was less than 0.1 nm and 0.15 nm for IL-2Rα in complex with wtIL-2 and M1, respectively, sharp jumps with RMSD values more than 0.3 nm were observed for the subunit in complex with M2. It shows that IL-2Rα face significant structural rearrangements to be able to stablish new interactions or loos interactions with M2, which is also evident in the RMSD plot of M2 in complex with IL-2Rαβγc (Fig 3D).

**Table 2. Main interactions observed between wtIL-2, M1, and M2 mutants with IL-2Rα During 100ns of MDs.**

| IL-2 Residues | IL-2Rα Residues | Distance | Specific Interactions | HB | Salt bridge | Surface Complementarity | Buried SASA |
|---|---|---|---|---|---|---|---|
| K35 | E1<br>D4 | 2.7 Å<br>4.0 Å | 1x hb, 1x salt bridge to E1 | 1 | 1 | 0.76 | 71.1% |
| R38 | F121<br>C3<br>H120<br>D4 | 3.0 Å<br>3.1 Å<br>3.5 Å<br>3.9 Å | 1x hb to F121 | 1 | 0 | 0.86 | 80.6% |
| T41 | N27<br>I118<br>H120 | 3.3 Å<br>3.9 Å<br>3.9 Å | 1x hb to N27 | 1 | 0 | 0.85 | 61.6% |
| F42 | H120<br>N27<br>L42 | 3.1 Å<br>3.6 Å<br>3.7 Å | 1x π-alkyl to L42 | 0 | 0 | 0.76 | 99.8% |
| K43 | R36<br>E29 | 2.7 Å<br>3.5 Å | 1x hb, 1x salt bridge to E29<br>1x hb to R36 | 2 | 1 | 0.44 | 53.0% |
| F44 | R36 | 3.5 Å | 1x π-alkyl R36 | 0 | 0 | 0.65 | 93.1% |
| Y45 | R35<br>R36 | 3.5 Å<br>3.5 Å | 1x π-alkyl, 1x π-cation to R35<br>1x π-alkyl to R36 | | | | |
| E61 | R35<br>K38 | 2.7 Å<br>3.6 Å | 1x hb, 1x salt bridge to R35 | 1 | 1 | 0.51 | 80.3% |
| E62 | R36 | 2.7 Å | 2x hb, 1x salt bridge to R36 | 2 | 1 | 0.50 | 97.0% |
| P65 | R36 | 3.3 Å | 1x π-alkyl to L42 | 0 | 0 | 0.79 | 98.7% |
| E68 | Y43<br>L42 | 2.6 Å<br>3.7 Å | 1x hb to Y43 | 1 | 0 | 0.86 | 60.1% |
| L72 | M25<br>Y43 | 3.7 Å<br>3.5 Å | 1x π-alkyl to Y43 | 0 | 0 | 0.44 | 63.8% |
| Y107 | S64<br>R35 | 2.9 Å<br>3.2 Å | 1x hb to S64 | 1 | 0 | 0.79 | 96.4% |
| E110 | R32 | 2.7 Å | 2x hb, 1x salt bridge to R32 | 2 | 1 | 0.86 | 39.0% |
| **M1 Residues** | **IL-2Rα Residues** | **Distance** | **Specific Interactions** | **HB** | **Salt bridge** | **Surface Complementarity** | **Buried SASA** |
| A42 | L42 | 4.0 Å | _ | 0 | 0 | 0.41 | 100% |
| E61 | S39<br>K38 | 2.7 Å<br>2.8 Å | 1x hb, 1x salt bridge to K38<br>1x hb to S39 | 2 | 1 | 0.71 | 71.9% |
| E62 | R36<br>S39<br>R35<br>K38 | 2.7 Å<br>2.8 Å<br>2.8 Å<br>3.9 Å | 1x hb to R35<br>2x hb, 1x salt bridge to R36<br>1x hb to S39 | 4 | 1 | 0.65 | 94.6% |
| A65 | G40<br>R36 | 3.6 Å<br>3.7 Å | _ | 0 | 0 | 0.81 | 98.0% |
| E68 | Y43 | 2.8 Å | 1x hb to Y43 | 1 | 0 | 0.29 | 38.6% |
| A72 | Y43 | 3.5 Å | _ | 0 | 0 | 0.87 | 60.8% |
| **M2 Residues** | **IL-2Rα Residues** | **Distance** | **Specific Interactions** | **HB** | **Salt bridge** | **Surface Complementarity** | **Buried SASA** |
| A42 | N27 | 3.7 Å | _ | 0 | 0 | 0.77 | 100.0% |
| A61 | S39<br>K38 | 3.0 Å<br>3.9 Å | 1x hb to S39 | 1 | 0 | 0.52 | 82.0% |
| E62 | R35 | 3.8 Å | 1x hb to R35 | 0 | 1 | 0.15 | 91.6% |
| E68 | Y43<br>S41 | 2.5 Å<br>2.7 Å | 1x hb to S41<br>1x hb to S43 | 2 | 0 | 0.66 | 61.7% |

Accordingly, the RMSD plots of IL-2 and M2 reach the plateau after 20 ns of simulations with RMSD values less than 0.1nm, but for that of M1, the plot reaches the steady-state after 30 ns of simulations with RMSD value of 0.1 nm, but starts to increase from nanosecond 80 and

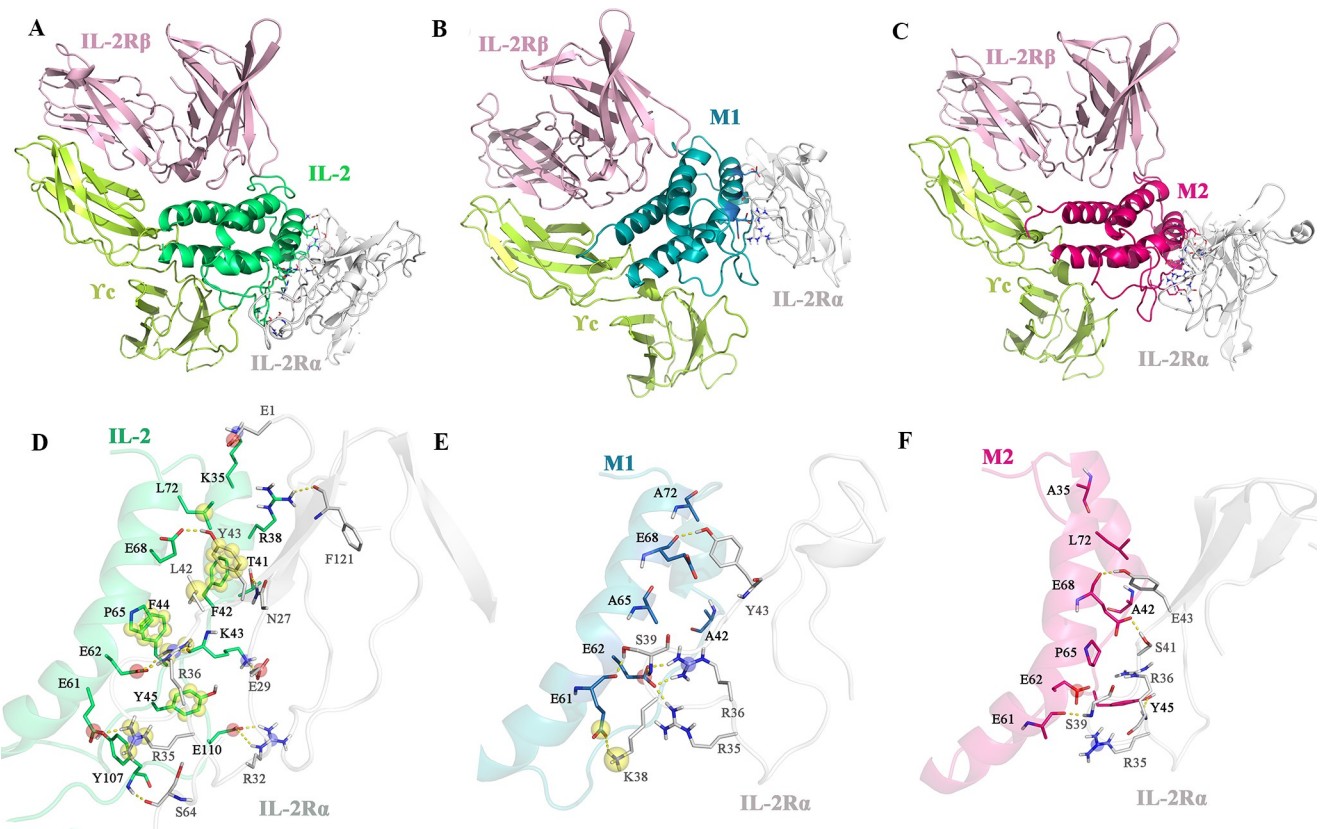

**Fig 4.** Interactions between **A.** wtIL-2 and IL-2Rαβγc, **B.** M1 and IL-2Rαβγc, and **C.** M2. IL-2Rαβγc after 100 ns of MDs, as well as the close view of **D.** wtIL-2 and IL-2Rα interactions, **E.** M1 and IL-2Rα and **F.** M2 and IL-2Rα. The hydrogen bonds are shown in yellow dashes. The red and blue spheres represent charging interactions (salt bridges), and the yellow spheres represent electrostatic (like π-π, alkyl-π and π-cation) interactions.

reach the plateau again in nanosecond 90 and stay steady for the rest of the simulation time. This shift in M1 RMSD plot can be referred to as the decrease in its affinity for IL-2Rβ and IL-2Rγc besides IL-2Rα, while the decline in M2 affinity for IL-2Rα enhances its interactions with the two other subunits (Fig 3).

The per-residue root mean-square fluctuation (RMSF) plots of IL-2Rβ and IL-2Rγc further confirms the stability of their interactions and conformation in complex with IL-2 and the mutants (Fig 3E and 3F). The RMSF plots of IL-2Rα in complex with M1 and M2 showed a non-significant increase (less than 0.1 nm) in the interacting residues fluctuations which is due to the loss of interactions with the mutants in comparison with IL-2 (Fig 3G). Also, the RMSF plots of IL-2, M1, and M2 show that the assigned mutations do not alter the native protein structure drastically. Since residues E61 to L72 are located in the B helix structure and substitution with alanine does not alter main chain interactions, notable fluctuations are avoided in the region. Also, fluctuations in F41A and F42A are because of their being involved in coil structure which is less than 0.1 nm (Fig 3H).

Interactions between IL-2, M1, and M2, with IL-2Rα were thoroughly studied with the application of Schrodinger software. According to Table 2 and Fig 4A, wtIL-2 interacts with IL-2Rα through hydrogen bond and salt bridge formation between residues K35, R38, T41, K43, E61, E62, E68, Y107, and E110 with residues E1, N27, E29, R32, R35, R36, Y43, S64, and F121, but no pi-stacking interactions are observed between the two proteins. The interactions are drastically reduced between M1 and M2 residues with IL-2Rα. While M1 forms hydrogen

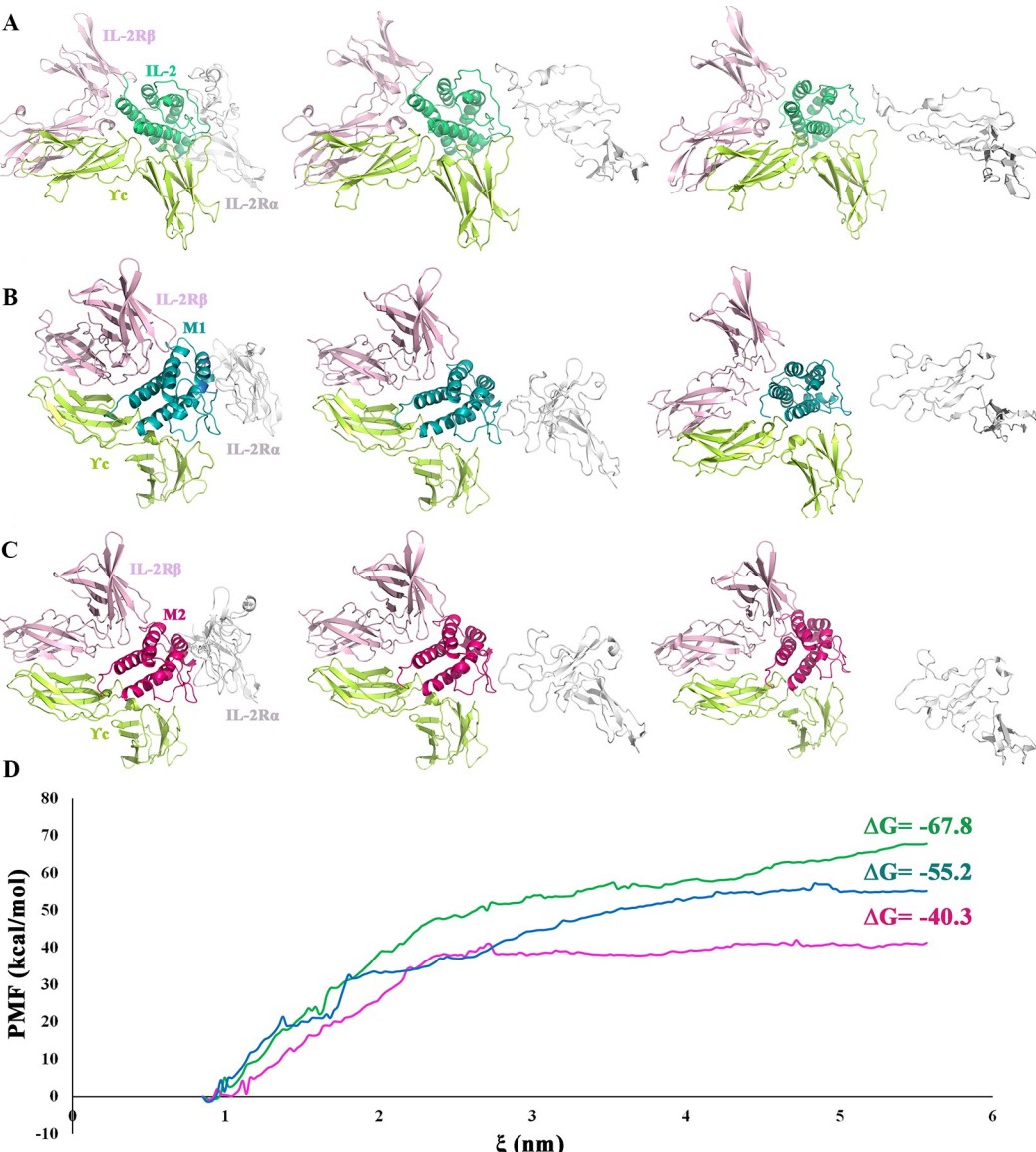

**Fig 5. Snapshots of umbrella sampling simulations of the target complex. A.** Native IL-2, **B.** Mutant 1and **C.** Mutant 2. **D.** Potentials of mean forces (PMF) for IL-2 and IL-2Rα (the native (green), mutant 1 (blue) and mutant 2 (magenta) proteins) as a function of the reaction coordinate (ξ).

bonds and salt bridges with R35, R36, K38, S39, and Y43, M2 only forms hydrogen bonds with residues R35, S39, S41, and S43 (Fig 4B and 4C).

Furthermore, the alternation in the number of hydrogen bonds between wt-IL2, M1, and M2 with IL-2Rβ, IL-2Rγc, and IL-2Rα were calculated during 100 ns of MDs (S3 Fig). Accordingly, while the number of hydrogen bonds between M2 and IL-2Rα is the least in comparison with those of wtIL-2 and M1, the interactions are almost constant for those of IL-2Rβ and IL-2Rγc in complex with M2. Although the focus of this study was to decrease the electrostatic interactions between IL2 and IL-2Rα, the stability in the number of hydrogen bonds between the mutants and IL-2Rβ and IL-2Rγc shows that the mutations did not disturb the target protein interactions with the IL2R. βγc.

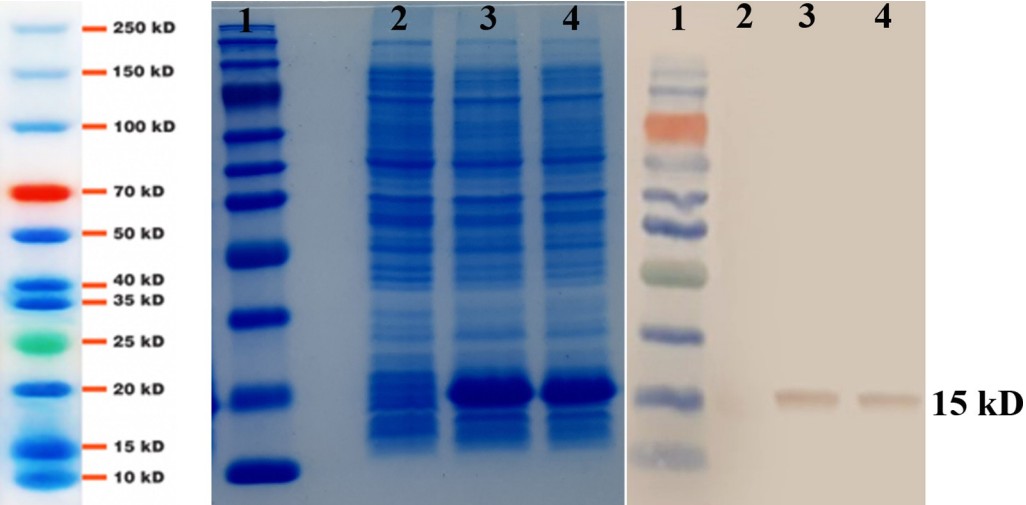

**Fig 6. SDS-PAGE and Western blotting analysis of the recombinant proteins.** Lane 1, protein marker (10–250 kDa). Whole lysate of BL21(DE3)-pET-22b without gene (lane 2), expressed wtIL-2 (lane3), mutant IL-2 (lane4).

**3.1.5. Umbrella sampling.** To calculate the binding free energy (ΔG) of the wtIL-2-ILR2αβγc, M1-hILR2αβγc, and M2-hILR2αβγc, IL-2Rα was assigned a constant velocity pulling procedure followed by umbrella sampling simulations. PMF curves were obtained from simulations of 23 windows for each system as a function of the distance between centers of mass (COMs) of wtIL-2, M1, and M2 and IL-2Rα using the weighted histogram analysis method (WHAM) (S4 Fig). According to Fig 5, wtIL-2 more strongly binds to IL-2Rα with a ΔG value of -67.8 kcal/mol in comparison with M1 and M2, indicating a stronger binding between wtIL-2 and IL-2Rα. Additionally, M2 with a ΔG value of -40.3 kcal/mol has the least affinity for IL-2Rα. Eventually, mutant 2 was selected for further experimental evaluations.

## 3.2. Experimental studies

**3.2.1. Protein expression and purification.** 12% SDS-PAGE and western blotting analyses confirmed the presence of approximately 15 kDa band for wild and M2 IL-2 in the crude and purified samples (Fig 6).

**3.2.2. Flow cytometry.** We used flow cytometry to evaluate the capacity of the new molecule to bind to the ConA activated PBMCs, which express high levels of trimeric IL-2R [29]. At equal concentration, the mean fluorescence intensity (MFI) was obtained 8.99 and 13.80 for the M2 and wtIL-2, respectively, showing the binding capacity of M2 was lower than wtIL-2 (Fig 7A). A competitive binding assay was also performed to determine whether the M2 can inhibit the binding of the anti-CD25 mAb, which competes with wtIL-2 (Fig 7B). The results indicated that the MFI value for the PBMCs which were preincubated with M2 was almost the same as the un-incubated cells (21.1 vs. 21.8, respectively); while, the pre-incubation of cells with wtIL-2 decreased the MFI to 10.2.

## 4. Discussions

Considered the first effective cancer immunotherapy, HD-IL-2 therapy has a response rate of approximately 20% in patients with metastatic melanoma and renal cell cancers. However, high dose IL-2 treatment resulted in an unwanted expansion of immunosuppressive Tregs through preferential binding of IL-2 to the high-affinity IL-2R, which have limited the efficacy

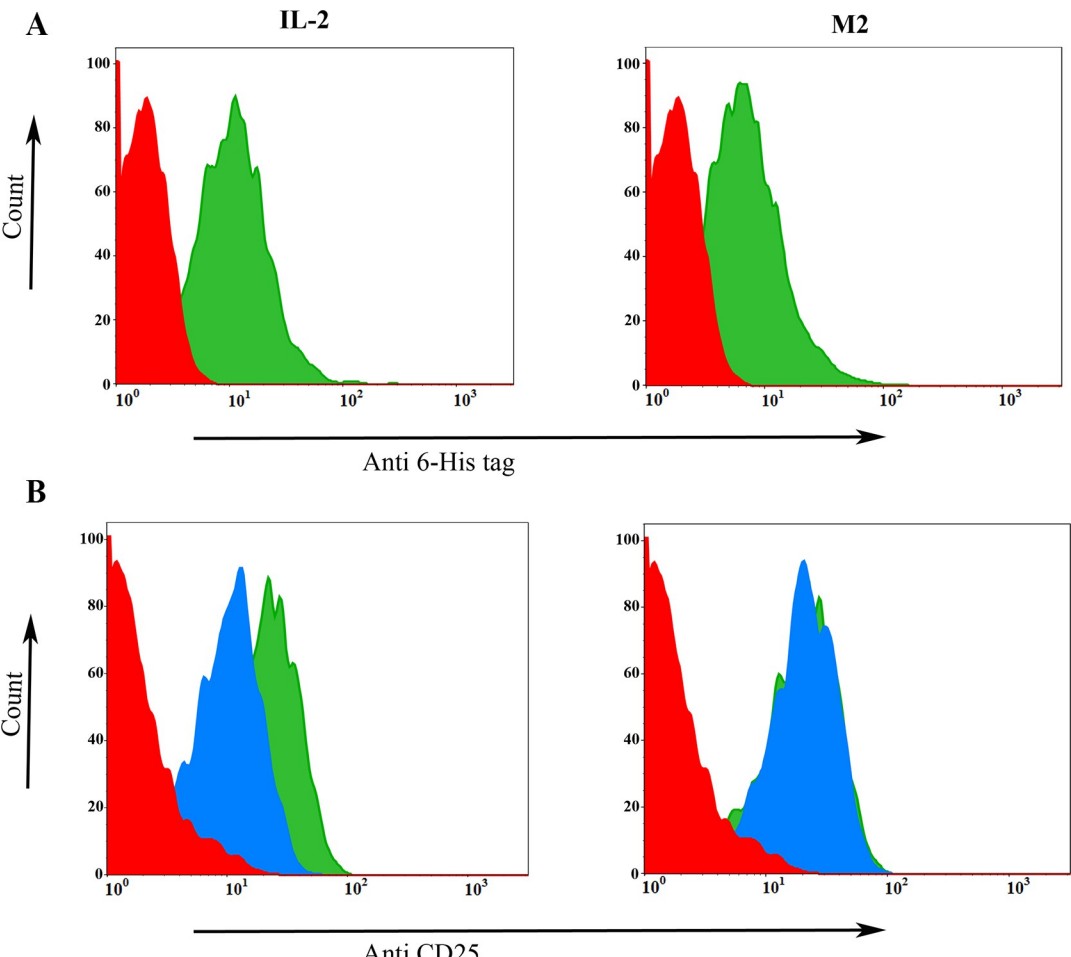

**Fig 7. Flow cytometry analysis showing the binding ability of IL-2 mutein to the conA activated PBMC. A.** Direct binding, red graph negative control, green graph wtIL-2 (left panel) and mutant IL-2 (right panel) bound to the cells and detected with rabbit anti-His-tag Ab and mouse anti-rabbit FITC-conjugated. **B.** Flow cytometry analysis showing the competitive assay. Negative control (red graph), cells labeled with anti-CD25 MAb-PE (green), and cells previously incubated with wtIL-2 (blue graph; left panel) or mutant IL-2 (blue graph; right panel) and then labeled with the anti-CD25 MAb-FITC.

and broad application of IL-2 in the clinic [30]. It is therefore of considerable interest to find methods that improve IL-2 as an antitumor agent. Introducing targeted mutations to modifying IL-2 function is an interesting way through which the subsequent immune response can be controlled to favor suppressive or cytotoxic responses. Accordingly, mutations that decrease the affinity or interrupt CD25 binding are typically employed in order to preferentially stimulate cytotoxic CD8[+] T and NK cells, while not affecting Tregs. Furthermore, the use of IL-2 muteins can disfavor contact with endothelial cells and significantly reduces the major side effect of HD-IL-2 immunotherapy, called vascular leak syndrome (VLS), which is caused by the direct binding of IL-2 to CD25[+] pulmonary endothelial cells [31,32]. Pulmonary edema is caused by direct binding of IL-2 with the functional form of IL-2 receptors (IL-2Rαβγc) on lung endothelial cells and blocking the a-chain leads to dramatically reduction of the pulmonary toxicity induced by IL-2 [3].

It has already been reported that IL-2 interactions are mediated by two hydrophobic patches around F42, Y45, and L72 residues, located on A-B loop and helix B, and mutations in these residues result in loss of a large part of the Van der Waal interaction surface [34,35].

Heaton *et al.* have shown that secondary cytokine production, another cause of endothelial damage, is dramatically reduced by R38A and F42K muteins, which preferentially binds with an intermediate-affinity [33,34]. Such IL-2 variants provide an effective, yet less toxic means of cancer immunotherapy. This property is important because the high toxicity is the main drawback of IL-2 therapy. Another IL-2 mutein was generated by substitution of alanine at residues of R38, F42, Y45, and E62, resulting in decreased affinity for CD25 without affecting normal binding with IL-2Rβγc [22]. The mutein inhibited the metastasis of the B16 melanoma variant MB16F0 and 3LL-D122 Lewis lung carcinoma in mice, while causing less toxic effects compared to wtIL-2 [3].

Following our *in-silico* simulations, two sets of triple mutations were assigned to wtIL-2 structure, both including F42A substitution located in the A-B loop. M1 also includes residues L72 and P65 that are parts of helix B. Mutations in F42A and P65A disturb the hydrophobic patch composed of aromatic rings shared by F42, P65, and Y45 residues (S5 Fig). Also, F42A and L72A substitutions eliminate the π-alkyl interactions with L42 and Y43 residues from IL-2Rα (Fig 4D and 4E). Due to the loss of hydrophobic ridges around residues F42 and Y45, supported by the aromatic-aromatic interactions by P65 (Figs 1F and S5), the A-B loop cannot enter into the grooves between IL-2Rα strands to stabilize the IL-2 and IL-2Rα close to each other. Eventually, other interactions like Y45, T41, and D35 interactions that are located in more flexible regions of IL-2 (A-B loop and α) are lost as well.

Besides F42A substitution in M2 that avoids hydrophobic interactions establishment, the mutant includes K35A and E61A substitutions, located in α and B helixes. Mutation in K35A disturbs hydrogen bond as well as salt bridge formation with residue E1 of IL-2Rα and E61A substitution eliminates attractive charge interactions with K38. The obtained results show that K35A, F42A, and E61A further decrease the affinity for IL-2Rα with a ΔG value of -40.3 kcal/mol in comparison to M1 and wtIL-2 with ΔG values of -55.2 and -67.8 kcal/mol, respectively. This energy difference in M1 and M2 can be attributed to the distribution of the mutations in the interacting regions. While mutations are assigned in the middle and top of the interacting region in M1, mutations in M2 are assigned throughout the region.

The performed *in vitro* experiments by flow cytometry indicated that the anti-CD25 mAb was able to bind to PBMC cells even after IL-2 mutant (M2) preincubation, therefore, the binding strength of the M2 to α-subunit is less than wtIL-2. Our results are in agreement with the previous investigation of the reduced capacity of the M2 compared to wtIL-2. Such mutations that severely reduce binding of α subunit, preferentially stimulate cytotoxic CD8[+] T and NK cells while decrease interaction with Tregs and endothelial cells that resulted in low toxicity and are more effective in immunity than wtIL-2 [3,7,34,35].

## 5. Conclusions

Due to reduction in IL-2 binding to its receptor on high affinity cells, it was decided to reduce the affinity of IL-2 to its α-subunit by inserting mutations. A computational evaluation methodology was designed and employed with the aid of docking, molecular dynamic simulations and umbrella sampling technique to evaluate and identify mutations that would reduce the affinity of IL-2 to its natural subunit of IL-2Rα. Accordingly, from the two triple mutants designed and studied, IL-2 mutant 2 (K35A, E61A, and F42A) showed noticeably a reduced affinity for IL2-Rα. The computationally obtained results were further evaluated and confirmed with the aid of flow cytometry technique. Our experimental findings showed that the affinity of M2 has significantly reduced to high-affinity ConA activated PBMCs relative to wtIL-2. Therefore, further studies in this field should be continued. Also, the obtained results demonstrate that computer-aided design of single-site amino acid mutations is an applicable

strategy to modulate binding between two proteins with an already highly optimized interface and affinity in the nanomolar range.

## Supporting information

**S1 Fig. A diagram of the whole computational study procedure.**
(TIF)

**S2 Fig.** Ramachandran plots of the final A. wtIL-2, B. Mutant 1, and C. Mutant 2, after 100 ns of MDs. Highly preferred, preferred and questionable observations are shown as green crosses, brown triangles, and red circles, respectively.
(TIF)

**S3 Fig.** The number of hydrogen bonds between wtIL-2 (green), M1 (blue) and M2 (magenta) with A. IL-2Rβ, B. IL-2Rγc, and C. IL-2Rα.
(TIF)

**S4 Fig.** All umbrella sampling window simulations for, A. IL-2Rα begins dissociation from wtIL-2 (green), B. IL-2Rα begins dissociation from M1 (blue) and C. IL-2Rα begins dissociation from M2 (magenta). D. The force on the spring over 500 ps of MDs for IL-2Rα begins dissociation from wtIL-2 (green), M1 (blue) and M2 (magenta).
(TIF)

**S5 Fig.** The close view of the A-B loop and the grooves between IL-2Rα strands, the hydrophobic ridges around residues F42, Y45, and L72 and their electrostatic interactions. The yellow shperes represent hydrophobic areas and the two end arrows represent electrostatic (like π-π, alkyl-π and π-cation) interactions.
(TIF)

**S1 Table. The dissociation constants (Kd) and binding affinities (ΔG) of wt-IL-2 and the variants (after the point mutation assignment) for IL-2Rα (calculated at 25˚C).**
(DOCX)

**S2 Table. The percent of the residues in the favored, allowed and disallowed regions of the Ramachandran plots of the studied variants after 60 ns, 80 ns, and 100 ns of MDs.**
(DOCX)

**S3 Table. The dissociation constants (Kd) and binding affinities (ΔG) of the wt-IL2, M1 and M2 for IL-2Rα, IL-2Rβ and IL-2Rγc (calculated at 25˚C).**
(DOCX)

## Acknowledgments

The authors wish to express their deep gratitude to all who provided support during the course of this research.

## Author Contributions

**Conceptualization:** Alireza Biglari, Mohammad Ali Shokrgozar, Sirous Zeinali, Yeganeh Talebkhan, Reza Ahangari Cohan.

**Investigation:** Arezoo Beig Parikhani, Rada Dehghan.

**Methodology:** Farhad Riazi Rad, Yeganeh Talebkhan.

**Project administration:** Kowsar Bagherzadeh.

**Software:** Arezoo Beig Parikhani, Kowsar Bagherzadeh.

**Supervision:** Sirous Zeinali, Soheila Ajdary, Mahdi Behdani.

**Writing – original draft:** Arezoo Beig Parikhani, Rada Dehghan.

**Writing – review & editing:** Kowsar Bagherzadeh, Yeganeh Talebkhan, Soheila Ajdary, Reza Ahangari Cohan, Mahdi Behdani.

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
