## [Decision Letter · Decision Letter 0]

26 Dec 2021

PONE-D-21-28229Human IL-2Rɑ Subunit Binding Modulation of IL-2 through a Decline in Electrostatic Interactions: A Computational and Experimental ApproachPLOS ONE

Dear Dr. Behdani,

Thank you for submitting your manuscript to PLOS ONE. Let me next say how sorry I am about the very extensive length of time your manuscript has been with PLOS ONE for evaluation. I sincerely hoped it did not affect the student from defending and graduating. There were several extenuating circumstances.  As you will see, there is one one external reviewer and myself. I have reviewed the manuscript and find the work interesting and suitable for PLOS ONE. However, I have one major concern with the manuscript that I believe can easily be remedy. The results second focuses thoroughly on the computational aspect of the study but failed to mention any of the 'wet biochemistry', that is, the binding studies with M1 and M2.  Those aspects were noted in the methods section but not in the results or discussion when the title noted it.  Also I do not know what M1 and M1 represent: what are the mutations: are they single or multi.     there were few typos in the text that I believe will be caught with the revision.      That said, we feel that the manuscript has merit but does not fully meet PLOS ONE’s publication criteria as it currently stands. Therefore, we invite you to submit a revised version of the manuscript that addresses the points raised during the review process. I promise to quickly evaluate the revision and make a decision.

We look forward to receiving your revised manuscript.

Kind regards,

Michael Massiah

Academic Editor

PLOS ONE

Journal Requirements:

This project was financially supported by Pasteur Institute of Iran. The authors wish to express their deep gratitude to all who provided support during the course of this research.

URL of each funder website

https://en.pasteur.ac.ir/

In your cover letter, please note whether your blot/gel image data are in Supporting Information or posted at a public data repository, provide the repository URL if relevant, and provide specific details as to which raw blot/gel images, if any, are not available. Email us at plosone@plos.org if you have any questions

Reviewers' comments:

Reviewer's Responses to Questions

**Comments to the Author**

1. Is the manuscript technically sound, and do the data support the conclusions?

Reviewer #1: Partly

2. Has the statistical analysis been performed appropriately and rigorously? 

Reviewer #1: N/A

3. Have the authors made all data underlying the findings in their manuscript fully available?

Reviewer #1: Yes

4. Is the manuscript presented in an intelligible fashion and written in standard English?

Reviewer #1: Yes

5. Review Comments to the Author

Reviewer #1: The authors used computational approach to study IL2 mutant to potential lower the binding affinity of IL2 to IL2R, trying to help design better IL2 drug to selectively activate effector T cells rather than Treg. The idea is very creative and the study is worth exploring. Just a few questions before it can be published.

1. IL2 binds to IL2Rabg and IL2Rbg, both provide activation signal to the downstream pathways, when the authors made the mutants. were the selected amino acids involved in the electrostatic interactions with IL-2Rα also involved in the interactions with IL2Rbg? Would the mutation affect IL2 - IL2Rbg interactions?

2. Since the goal of IL2 mutein was to avoid Treg expansion, an in vitro experiments would be very helpful to provide further evidence of the mutants that can activate effector T cells only

3. A diagram of how the authors performed umbrella sampling would be very helpful

6. PLOS authors have the option to publish the peer review history of their article (what does this mean?). If published, this will include your full peer review and any attached files.

Reviewer #1: No

---

## [Author Response · Author response to Decision Letter 0]

12 Jan 2022

Dear editor,

Please find enclosed the revised version of our manuscript “Human IL-2Rɑ Subunit Binding Modulation of IL-2 through a Decline in Electrostatic Interactions: A Computational and Experimental Approach” by Arezoo Beig Parikhani et al., which we submit for publication in the PLOSONE.

First, we would like to thank the editor for giving us the opportunity to submit a revised version and the reviewers for their critical but constructive comments on the first version. There have been remarks on the organization of the manuscript and we agree that it could have been done better. Changes to the manuscript are done with track change in text. What follows is a point-to-point explanation on how we answered the suggestions and requested amendments:

Reviewers' comments:

Academic editor

I have one major concern with the manuscript that I believe can easily be remedy. The results second focuses thoroughly on the computational aspect of the study but failed to mention any of the 'wet biochemistry', that is, the binding studies with M1 and M2. Those aspects were noted in the methods section but not in the results or discussion when the title noted it. 

Response: As mentioned in the text of the manuscript, initially insilico studies were performed on M1 and M2 mutant forms and based on the obtained results, M2 mutant was selected for laboratory studies by flow cytometry which represented in experimental section. 

Also, I do not know what M1 and M1 represent: what are the mutations: are they single or multi.

Response: According to the abbreviation list M1 and M2 represent the two mutant forms of human interleukin-2 (Mutant numbers 1 and 2). The triple mutations generated in these two variants have been mentioned in subheading “computational studies” of the result section in which M1 possessed F42A, P65A, and L67A mutations while K35A, F42A, and E61A mutations were selected for M2 molecule. These two variants were further studied and compared with wild type IL-2 molecule.

There were few typos in the text that I believe will be caught with the revision.

Response: Thanks for your attention. The whole text has been revised.

Journal Requirements:

1- Please ensure that your manuscript meets PLOS ONE's style requirements, including those for file naming.

Response: The manuscript has been fully reviewed according to PLOS ONE's style requirements.

2- Please note that funding information should not appear in the Acknowledgments section or other areas of your manuscript. We will only publish funding information present in the Funding Statement section of the online submission form. Please remove any funding-related text from the manuscript and let us know how you would like to update your Funding Statement. Please include your amended statements within your cover letter; we will change the online submission form on your behalf.

Response: Done.

3- PLOS ONE now requires that authors provide the original uncropped and unadjusted images underlying all blot or gel results reported in a submission’s figures or Supporting Information files. In your cover letter, please note whether your blot/gel image data are in Supporting Information or posted at a public data repository, provide the repository URL if relevant, and provide specific details as to which raw blot/gel images, if any, are not available. Email us at plosone@plos.org if you have any questions

Response: Done.

4- Please review your reference list to ensure that it is complete and correct.

Response: The references have been rechecked thoroughly.

Reviewer #1: 

The authors used computational approach to study IL-2 mutant to potential lower the binding affinity of IL-2 to IL-2R, trying to help design better IL-2 drug to selectively activate effector T cells rather than Treg. The idea is very creative and the study is worth exploring. Just a few questions before it can be published.

1- IL-2 binds to IL-2Rabg and IL-2Rbg, both provide activation signal to the downstream pathways, when the authors made the mutants. Were the selected amino acids involved in the electrostatic interactions with IL-2Rα also involved in the interactions with IL2Rbg? Would the mutation affect IL2 - IL2Rbg interactions?

Response: Thanks for the question. In order to confirm that decreased affinity is due to the decline in target residual interactions with IL-2Rα, the affinity of wtIL-2 and the mutant forms towards IL-2Rβ and IL-2Rγc were also studied by PRODIGY (S3 table). Although the affinity of M2 to IL-2Rα decreased, an increased affinity was observed towards IL-2Rβ. The assigned mutations affected the affinity of M2 protein to the IL-2Rγc. Since the IL-2Rγc binding site is very close to that of IL-2Rα, mutations in the latter would probably induce conformational changes in the former site. Therefore, docking studies can assume that M2 mutations do not have any significant effect on IL-2 interaction with IL-2Rβ and IL-2γc. 

In addition, based on MD studies the alternation in the number of hydrogen bonds between wt-IL2, M1, and M2 with IL-2Rβ, IL-2Rγc, and IL-2Rα were calculated during 100 ns (S2 Fig). In the case of M2 molecule, while the number of hydrogen bonds between M2 and IL-2Rα was the least in comparison with other IL-2 proteins, the interactions were almost constant for IL-2Rβ and IL-2Rγc confirming the neutral effect of the mutations on main interactions between M2 and the two mentioned receptors.

It should also be mentioned that the three IL-2 receptors binding sites are distinct and if the mutations affect M2 interactions with IL-2Rβ and IL-2Rγc (while not significantly according to our studies, it is not directly but inducive.

2- Since the goal of IL-2 mutein was to avoid Treg expansion, an in vitro experiment would be very helpful to provide further evidence of the mutants that can activate effector T cells only.

Response: Good comment. Since the aim of the present study was to computationally investigate the effect of triple mutations on binding of IL-2 to the IL-2Rα receptor, the only in vitro experiment carried on was flow cytometry analysis. Further experiments are going to perform to study the effects on Tregs in in vitro and in vivo conditions which will be shown in ongoing manuscript. 

3- A diagram of how the authors performed umbrella sampling would be very helpful.

Response: Thanks for the suggestion. A diagram of the whole computational study including umbrella sampling has been added to supplementary information file.

---

## [Editor Report · Decision Letter 1]

9 Feb 2022

Human IL-2Rɑ Subunit Binding Modulation of IL-2 through a Decline in Electrostatic Interactions: A Computational and Experimental Approach

PONE-D-21-28229R1

Dear Dr. Behdani,

We’re pleased to inform you that your manuscript has been judged scientifically suitable for publication and will be formally accepted for publication once it meets all outstanding technical requirements.

Kind regards,

Michael Massiah

Academic Editor

PLOS ONE
---

## [Editor Report · Acceptance letter]

16 Feb 2022

PONE-D-21-28229R1 

Human IL-2Rɑ Subunit Binding Modulation of IL-2 through a Decline in Electrostatic Interactions: A Computational and Experimental Approach 

Dear Dr. Behdani:

I'm pleased to inform you that your manuscript has been deemed suitable for publication in PLOS ONE. Congratulations! Your manuscript is now with our production department. 

Kind regards, 

on behalf of

Dr. Michael Massiah 

Academic Editor

PLOS ONE